# Connecting Concept Convexity and Human-Machine Alignment in Deep Neural Networks

Teresa Dorszewski[*,1], Lenka Tětková[*,1], Lorenz Linhardt[2,3], and Lars Kai Hansen[1]

[1]Section for Cognitive Systems, DTU Compute, Technical University of Denmark
[2]Machine Learning Group, Technische Universität Berlin
[3]Berlin Institute for the Foundations of Learning and Data – BIFOLD

## Abstract

Understanding how neural networks align with human cognitive processes is a crucial step toward developing more interpretable and reliable AI systems. Motivated by theories of human cognition, this study examines the relationship between *convexity* in neural network representations and *human-machine alignment* based on behavioral data. We identify a correlation between these two dimensions in pretrained and fine-tuned vision transformer models. Our findings suggest that the convex regions formed in latent spaces of neural networks to some extent align with human-defined categories and reflect the similarity relations humans use in cognitive tasks. While optimizing for alignment generally enhances convexity, increasing convexity through fine-tuning yields inconsistent effects on alignment, which suggests a complex relationship between the two. This study presents a first step toward understanding the relationship between the convexity of latent representations and human-machine alignment.

## 1 Introduction

As machine learning models are increasingly integrated into various aspects of daily life, understanding and improving human-machine alignment holds the promise of a safe, reliable and fair deployment of these technologies, ensuring that they operate in ways that are ethical, transparent, and beneficial to society [1]. Subsequently, we see a rising academic interest in the alignment of humans and machine learning models [2–7], as well as the agreement between different alignment measures [8, 9]. This paper explores the connection between two quantities characterizing the structure of neural network representations inspired by cognitive science: a theory-based measure of convex regions in neural networks [10], inspired by Gärdenfors' theory of conceptual spaces [11], and an empirical measure of human-machine alignment based on a triplet odd-one-out task [7].

Gärdenfors' theory posits that human cognition can be understood in terms of geometric struc-

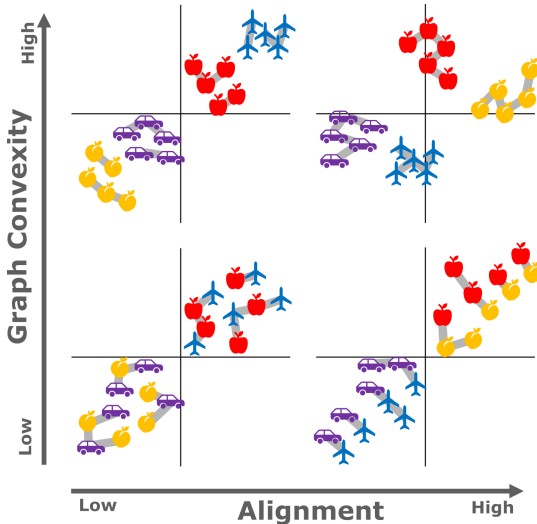

**Figure 1.** Toy example with four potential cases of alignment vs. convexity in representation space. Apples and peaches (or cars and airplanes) should have high cosine similarity to achieve high alignment with humans. All objects should be neighboring objects (gray connection) of the same type for high convexity. In principle, high convexity does not necessarily imply high alignment and vice versa. We investigate which of the scenarios best reflects the representation structure of trained neural networks.

tures, where concepts are represented as convex regions. Convexity is argued to assist generalization and communication of representations between humans [11]. Tětková et al. [10] introduced a framework to measure the convexity of conceptual regions in neural networks and discovered pervasive convexity across many data domains and models. They find that greater convexity leads to better generalization, which aligns with human cognition theories. In this work, we ask the fundamental question of whether these convex regions align with concepts used in human decision-making.

Human-machine alignment can be measured in several ways [8], one of which is by comparison with humans' relative similarity judgments (e.g. [4, 7]). This may involve an odd-one-out image triplet task (*"Which of three images is the most different?"*) using model representations and comparing the outcomes with the response of humans on the same

---

* Equal contribution. {tksc,lenhy}@dtu.dk

Proceedings of the 6th Northern Lights Deep Learning Conference (NLDL), PMLR 265, 2025.

task [7]. We chose this alignment metric as it can be easily evaluated for any neural network and additionally, methods for increasing this form of alignment exist [12].

We experimentally investigate the relationship between convex conceptual regions in neural network representations and alignment with human similarity judgments. We address two primary research questions:

(1) **Correlation between Convexity and Human-Machine Alignment**: To what extent are convexity and human-machine alignment correlated in real-world models? Figure 1 illustrates potential scenarios of high and low levels of convexity and alignment. In particular, high convexity does not necessarily imply high alignment or vice versa. Yet, in non-pathological real-world scenarios, there may be a relationship between alignment and convexity. We perform extensive experiments suggesting that these two quantities are indeed correlated.

(2) **Impact of Improving Human-Machine Alignment on Convexity**: Does increasing human-machine alignment lead to increased convexity? To understand whether the connection between alignment and convexity is causal, we increase human-machine alignment by applying the latent space transformations developed by Muttenthaler et al. [7, 12] and examine the effect on the convexity of conceptual regions. Additionally, we look at the effect of fine-tuning on convexity and alignment. Our findings indicate a complex relationship, showing an increase in convexity when optimizing pretrained models for alignment, but inconsistent effects on alignment during fine-tuning.

The motivation of this work is two-fold. First, both convexity and the investigated measure of human-machine alignment are proposed as indicators of human-likeness in neural network representations. If these measures were to exhibit a strong positive correlation, it could suggest a shared underlying phenomenon that needs to be better understood. This, in turn, could provide valuable insights into how neural networks represent inputs, which has practical implications for fields such as mechanistic interpretability, with the goal of enhancing neural network robustness and safety. Conversely, if convexity and the alignment measure show no or a negative correlation, it might indicate that one of these measures is a flawed indicator of human-likeness.

Second, both convexity and the alignment measure have been shown to improve task performance in certain contexts[10, 12]. Examining the interaction between these two measures could reveal whether their performance improvements can be expected to be cumulative - suggesting that networks should be optimized for both - or whether they stem from the same underlying mechanism.

## 2 Background

In this work, we investigate the connection of two already developed measures, namely the *graph convexity score* [10] and the *odd-one-out accuracy* [7].

### 2.1 Graph Convexity

The graph convexity score measures the convexity of concepts in latent spaces of neural networks [10]. It extends the common definition of convexity for Euclidean spaces to curved manifolds and utilizes sampled data. Graph convexity is defined as follows [10, 13]:

**Definition 1** (Graph Convexity). *Let $(V, E)$ be a graph and $A \subseteq V$. $A$ is convex if for all pairs $x, y \in A$, there exists a shortest path $P = (x = v_0, v_1, v_2, \ldots, v_{n-1}, y = v_n)$ and $\forall i \in \{0, \ldots, n\}$: $v_i \in A$.*

The graph convexity score measures the ratio of points within the same class that constitute the shortest path connecting any two points of the same class. It is constructed in the following way: first, we extract the latent representations of all data points in a given layer $(V)$. Then, we construct a nearest neighbor graph (with N=10) with the Euclidean distance measure. Next, for each pair of data points that belong to the same class $(x, y \in A)$, we find the shortest path $(P)$ in the neighbor graph and calculate the proportion of points along these paths that belong to the same class. The average of these proportions across all pairs is called the graph convexity score. See Appendix A.2 for a pseudo-code procedure for calculating the graph convexity score.

### 2.2 Human-Machine Alignment

As a measure of human-machine alignment, we use the triplet odd-one-out accuracy (OOOA) [7]. The underlying triplet task is based on the THINGS dataset [14], which contains images of natural objects. From this dataset, image triplets have been sampled, and human judgments of which of each triplet's images is the odd one out have been collected [15]. The alignment between humans and a neural network on the THINGS triplet task is measured by how well the human-designated odd one out can be directly identified using cosine-similarity of the images in latent space. For this, we first construct a similarity matrix $\boldsymbol{S} \in \mathbb{R}^{3 \times 3}$ where $S_{i,j} := \boldsymbol{x}_i^\top \boldsymbol{x}_j / (\|\boldsymbol{x}_i\|_2 \|\boldsymbol{x}_j\|_2)$, the cosine similarity between a pair of representations, for the representations $\boldsymbol{x}_1, \boldsymbol{x}_2$ and $\boldsymbol{x}_3$ of images of a given triplet. We center all representations by subtracting the mean representation over the THINGS dataset and divide them by the standard deviation. We identify the closest pair of images in the triplet as $\arg\max_{i,i<j} S_{i,j}$.

The remaining image is the odd one out. The odd-one-out accuracy is defined as the proportion of matching responses between humans and the model. Important to note is that the observed agreement between humans is 67.22% and this value thus upper bounds the expected OOOA values [15].

To improve human-machine alignment (i.e. OOOA), we use the naive transform defined by Muttenthaler et al. [12]. It affinely transforms the latent space to maximize the alignment between human similarity judgments and the network's representations. This transformation consists of a square matrix $\boldsymbol{W}$ and bias $\boldsymbol{b}$ obtained as the solution to

$$\underset{\boldsymbol{W},\boldsymbol{b}}{\arg\min} \ \mathcal{L}_{\text{global}}\left(\boldsymbol{Z}\right) + \lambda\|\boldsymbol{W}\|_{\boldsymbol{F}}^{2}, \qquad (1)$$

where $Z_{ij} = \left(\boldsymbol{W}\boldsymbol{x}_i + \boldsymbol{b}\right)^{\top}\left(\boldsymbol{W}\boldsymbol{x}_j + \boldsymbol{b}\right)$ where $\{i,j\}$ index any pair of representations in the dataset. The log-likelihood is defined as [16]:

$$\mathcal{L}_{\text{global}}\left(\boldsymbol{Z}\right) := -\frac{1}{n}\sum_{s=1}^{n}\log\underbrace{p\left(\{a_s,b_s\} \mid \{i_s,j_s,k_s\},\boldsymbol{Z}\right)}_{\text{odd-one-out prediction}}.$$
$$(2)$$

Here, $n$ is the number of triplets, and the probability of a pair, $\{a,b\}$ of the triplet $\{i,j,k\}$, being most similar, is given by the softmax over the representation similarities in $\boldsymbol{Z}$.

## 3 Methods

### 3.1 Data

For all experiments, we used the THINGS dataset [14] and its attached human triplet responses [15]. The THINGS dataset consists of over 26.000 images of 1854 classes of natural objects, which can be grouped into 27 high-level categories (determined by human crowd-sourcing), which we will refer to as *superclasses*. The dataset features over 4.7 million human triplet odd-one-out responses, based on one image per class.

We used the same images chosen for the triplet experiment to determine the OOOA and 100 randomly chosen images per superclass from the THINGS dataset to determine the graph convexity score.

### 3.2 Models

Although convexity and human-machine alignment are domain-independent concepts, we restricted ourselves to vision models, as the THINGS dataset used to measure alignment in prior work [7, 12, 17] consists of images. We compared three different transformer-based vision models: Vision Transformers (ViT) [18], Bidirectional Encoder representation from Image Transformers (BEiT) [19], and

data2vec [20]. For each model, we compared a base and a large architecture and both in their pretrained and fine-tuned (for ImageNet-1k [21] classification) version.[1] The models consist of a feature extractor followed by 12 or 24 transformer layers [22] with an embedding size of 768 or 1024 respectively. While ViTs were pretrained for classification, BEiT and data2vec were pretrained using a self-supervised objective, where the goal was to predict masked-out input (data2vec) or representation tokens (BEiT). All models were pretrained on ImageNet-21k [21].

### 3.3 Experiments

**Correlation between Convexity and Human-Machine Alignment.** To answer whether convexity and human-machine alignment are related, we perform a correlation analysis between the graph convexity and the odd-one-out-accuracy (OOOA). For this, we extracted the latent representations of the images used for the triplet after each transformer layer and measured the OOOA of the centered representations (as described in subsection 2.2). For the convexity analysis, we extracted the latent representations of 100 images for each superclass of the THINGS dataset after each transformer layer. We averaged the convexity scores of all classes to get one score per layer. We performed a correlation analysis between the two scores using Pearson's R on a layer-wise basis across all models.

**Impact of Improving Human-Machine Alignment on Convexity.** After addressing the correlation between the human-machine alignment and the convexity score, we ask how the optimization of one score will influence the other. To answer this question, we trained the naive transform described in subsection 2.2 to increase the OOOA of the latent representations after the first, middle, and last transformer layers. We then applied the transform to the latent representations of the remaining images and evaluated the graph convexity of these transformed latent representations. Furthermore, we examined the impact of fine-tuning on both the convexity and OOOA.

## 4 Results & Discussion

### 4.1 Correlation between Convexity and Human-Machine Alignment

First, we examine the progression of the convexity score and the OOOA across layers of different models and investigate the correlation between the two.

The graph convexity increases consistently for the first half of all evaluated models and then continues to increase for fine-tuned models but plateaus or

---

[1] All models were obtained from huggingface.co, for exact models see Table A.1.

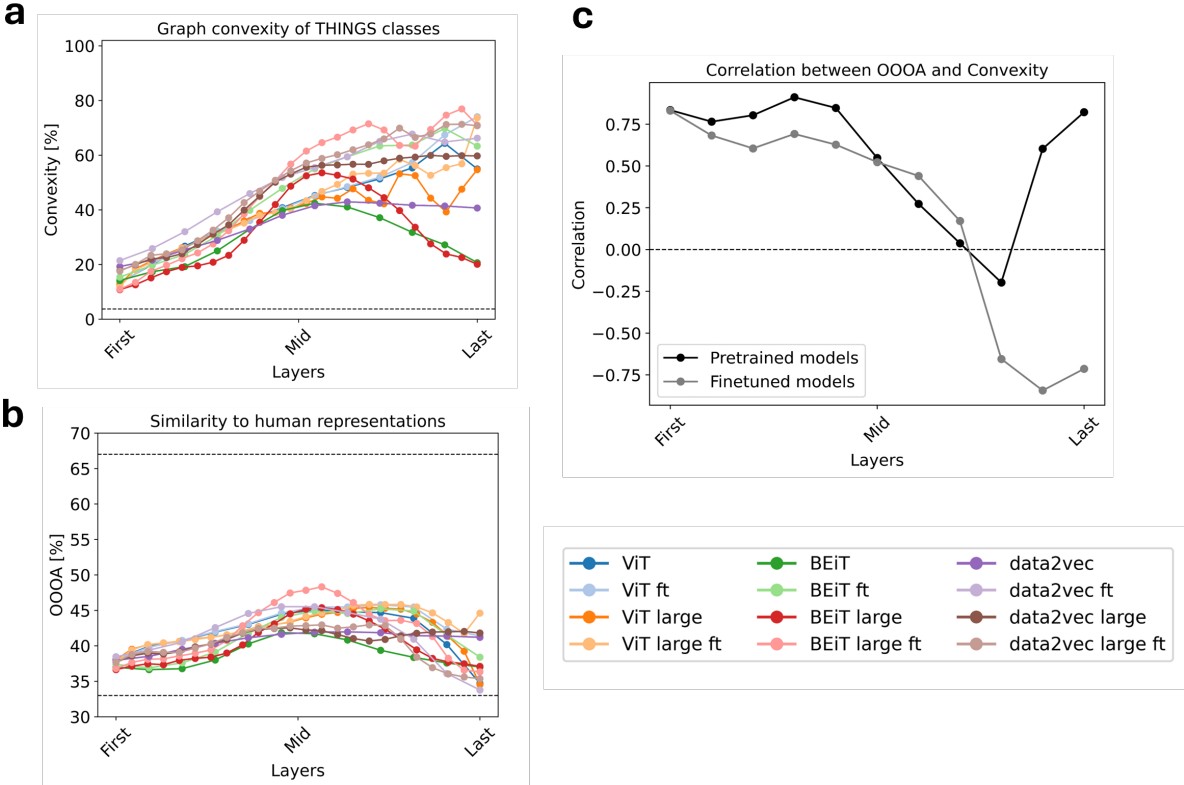

**Figure 2.** Correlation between convexity and human-machine alignment for ViT [18], BEiT [19] and data2vec [20] (all in base and large architecture, pretrained and fine-tuned (ft)) across all layers: **a)** Graph convexity of the THINGS superclasses across all models. Convexity increases steadily for fine-tuned models and peaks in the middle layers for pretrained models. Dotted line indicates the lower bound (random labeling). **b)** Human-machine alignment measured by OOOA across all models. OOOA peaks in the middle layers. Dotted lines indicate the lower bound (chance level) and upper bound (inter-human consistency level). **c)** Correlation between convexity and OOOA for pretrained and fine-tuned models respectively. The correlation is calculated across all models per layer (for large models we take every second layer, corresponding to the same relative position in the network as in the small networks). There is a strong positive correlation in the first half of the model. Fine-tuned models show a strong negative correlation in late layers.

decreases for pretrained models (Figure 2a). The convexity is generally higher in fine-tuned models, consistent with prior work [10]. The bell curve pattern observed in pretrained unsupervised models (BEiT and data2vec) may be attributed to their training objective, which employs a reconstruction loss. As the models learn to rebuild the masked input data, the later layers increasingly build latent representations akin to those in the earlier layers. Thus, intermediate layers may be the ones with the highest level of conceptual abstraction. A similar phenomenon has been observed previously in speech representation models [23–25].

The OOOA also follows a bell-shaped curve, peaking around the middle layers of the model and decreasing again toward the end (Figure 2b). Current human-machine alignment analyses focus mainly on the last or penultimate layer, as this bell curve was not observed in the supervised and contrastive models examined in previous work [7]. Our results suggest that the intermediate layers of the studied vision transformer models are better aligned with the human similarity space than the last layers. This is in line with observations on denoising diffusion models [17] and should motivate a careful approach for layer selection in future research on the alignment of neural networks. In contrast to our results on convexity, the bell curve can be observed for alignment in pretrained and fine-tuned models alike, albeit the latter generally achieve a higher maximum OOOA. This indicates that fine-tuning affects OOOA and convexity differently.

The correlation between the OOOA and convexity follows two main trends (see Figure 2c). In the first half of the models, both the convexity and the OOOA increase and are highly correlated, whereas the correlation decreases in the second half of the models. There is a considerable difference between pretrained and fine-tuned models in the later layers: for pretrained models, there is a mostly positive correlation that increases again in the last layers, while there is a strong negative correlation in late

layers for fine-tuned models.

In general, we observe a high correlation between human-machine alignment (OOOA) and the convexity of the 27 high-level concepts in many scenarios. This suggests that the networks form convex regions in their latent spaces that align with human-defined categories, which is also consistent with the similarity relations humans use to solve the triplet task. Especially in the first half of the models, which we interpret as the concept built-up phase, the correlation is strong. For the later layers, which can be viewed as processing the built-up concepts, e.g. for the purpose of classification, the correlation decreases and even becomes negative. This potentially indicates that the models initially learn to distinguish concepts, where human-like categorization and similarity judgments are beneficial, especially in unsupervised settings. In the subsequent phase and during fine-tuning, the similarity between classes seems to become less critical. For reconstruction-based models, the reason may be that high-level information captured during the first phase may be decomposed into low-level features in later layers to solve the task (i.e. pretraining objective). For classification models, the separation of classes for classification might not necessitate retaining the similarity structure across concepts.

## 4.2 Impact of Improving Human-Machine Alignment on Convexity

We investigate how increasing human-machine alignment influences convexity. Using the naive transform [12], the OOOA of all models improves significantly – on average by 13.7% for pretrained models and by 13.0% for fine-tuned models. We find that improving the OOOA also leads to higher convexity of representations of the THINGS superclasses in pretrained models; the convexity of the transformed latent representations increases for all but one model by on average 3.1% (see Table 1). This relation is strongest in the last layers, but also holds in other layers of the models (see Table A.2 and A.3 in the appendix).

For fine-tuned models, this relation does not hold consistently (see Table A.4-A.5 in the appendix). For most models, convexity decreases marginally, which is in line with the negative correlation scores in the late layers. We hypothesize that the training strategy (supervised vs. self-supervised) also has an impact on the effect of alignment optimization on convexity (preliminary results in Appendix A.4).

Increasing convexity does not necessarily lead to higher alignment, as observed in the fine-tuned models. Although classification performance and convexity increase during fine-tuning, this does not have a consistent effect on human-machine alignment. While fine-tuning increases OOOA for ViT, it de-

**Table 1.** Change in OOOA and convexity after naive transform (transf.) in the last layer. Standard error of the mean (SEM) for convexity ±0.1. Overall, the convexity increases when optimizing for human-machine alignment (OOOA).

| Model | OOOA [%] | | Convexity [%] | |
|---|---|---|---|---|
| | orig. | transf. | orig. | transf. |
| ViT base | 34.7 | +15.4 | 56.5 | +1.7 |
| ViT large | 34.5 | +13.8 | 58.7 | -1.3 |
| BEiT base | 37.0 | +13.1 | 20.5 | +5.0 |
| BEiT large | 37.0 | +12.4 | 20.2 | +4.0 |
| data2vec base | 41.2 | +13.6 | 40.2 | +3.6 |
| data2vec large | 41.8 | +13.6 | 57.7 | +5.7 |
| avg. change ($\pm std$) | $+13.7 \pm 1.0$ | | $+3.1 \pm 2.5$ | |

**Table 2.** Change in convexity and OOOA after fine-tuning (ft.) in the last layer. SEM for convexity ±0.1. Convexity increases significantly for all models. Changes in OOOA are not consistent across models.

| Model | Convexity [%] | | OOOA [%] | |
|---|---|---|---|---|
| | orig. | ft. | orig. | ft. |
| ViT base | 56.5 | +16 | 34.7 | +6.6 |
| ViT large | 58.7 | +9.6 | 34.5 | +10.0 |
| BEiT base | 20.5 | +44.3 | 37.0 | +1.3 |
| BEiT large | 20.2 | +51.5 | 37.0 | -0.7 |
| data2vec base | 40.2 | +26.1 | 41.2 | -7.4 |
| data2vec large | 57.7 | +14.1 | 41.8 | -6.4 |
| avg. change ($\pm std$) | $+26.9 \pm 17.3$ | | $+0.58 \pm 6.9$ | |

creases OOOA for data2vec and has only a marginal impact on OOOA for BEiT (see Table 2). This indicates a complex interplay between alignment and convexity, which potentially depends on factors such as training objectives as we also observed above.

Another promising direction for future research is to develop methods to increase convexity (and performance) that also consider human-machine alignment. Positive relations between human-machine alignment and performance/robustness have been previously found [12, 26], although alignment is not a sufficient condition for high performance and optimizing solely for alignment can worsen performance by unlearning properties of representations that are not necessary to achieve high OOOA.

## 5 Conclusion

We presented first evidence of a relationship between convexity in neural network representations and empirical human-machine alignment. Our findings indicate a significant correlation between these two measures in some scenarios and that the intervention to promote alignment can also increase convexity. The correlation suggests that neural networks form convex regions in their latent spaces that

to some extent align with human-defined categories and reflect the similarity relations humans use in tasks such as the odd-one-out triplet task. In the studied models, we find the highest human-machine alignment in the middle layers, which could help inform future research.

We observed that optimizing for human-machine alignment through latent space transformations not only increases the odd-one-out accuracy (OOOA) but can also increase the convexity of the representations. This intervention effect underscores the potential for optimizing models to be both more aligned with human cognition and more effective in their performance.

Our results highlight that in the higher layers, convexity and alignment can exhibit a strong positive or a strong negative correlation. This indicates a complex interplay between alignment and convexity that may hinge on factors such as training objective, architecture, or training data. Further research is warranted to explore under which conditions convexity and human-machine alignment align.

Overall, our study offers first insights into the relationship between the convexity of latent representations and their alignment with human similarity judgments, further connecting cognitive science and deep learning research. Future work should investigate the causal relationships between alignment and convexity and explore new techniques to enhance both simultaneously, potentially leading to more aligned and generalizing models.

## Acknowledgments

This work was supported by the Novo Nordisk Foundation grant NNF22OC0076907 "Cognitive spaces - Next generation explainability", the DIREC Bridge project Deep Learning and Automation of Imaging-Based Quality of Seeds and Grains, Innovation Fund Denmark grant number 9142-00001B and by the Pioneer Centre for AI, DNRF grant number P1. LL gratefully acknowledges funding from the German Federal Ministry of Education and Research under the grant BIFOLD24.

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

# A  Appendix

## A.1  Implementation details

Code for the convexity analysis is available at https://github.com/LenkaTetkova/Convexity-of-representations.

Code for the human-machine alignment analysis is available at https://github.com/LukasMut/gLocal and https://vicco-group.github.io/thingsvision/Alignment.html.

All investigated models were obtained from huggingface.co, the exact models are in Table A.1.

We extract the representations after each transformer block, which have the dimensions of $768\times197$ for base models and $1024\times197$ for large models. For ViT we use the first embedding patch (the cls token, corresponding to the first row of length 768/1024), which represents the class token of the embedding, which is later used by the classifier head. For the other models, we average across the rest of the patch embeddings (resulting in a vector of length 768), as this is used by the classifier head.

## A.2  Graph convexity score procedure

---
**Algorithm A.1** Graph Convexity Score

---
1: $G = (V, E)$, $N$-nearest neighbor graph for $V$ latent space points with edges $E$
2: $convexity\_scores \leftarrow []$
3: **for** each class $A \subseteq V$ **do**
4:     **for** each pair $x, y \in A, x \neq y$ **do**
5:         **if** $\nexists$ shortest path between $x$ and $y$ **then**
6:             $path\_convexity \leftarrow 0$
7:         **else**
8:             $P = (v_0 = x, v_1, \ldots, v_n = y)$, the shortest path in between $x$ and $y$ in $G$
9:             **if** $n = 1$ **then**
10:                $path\_convexity \leftarrow 1$
11:             **else**
12:                $path\_convexity \leftarrow \frac{|\{v_i \in A | 0 < i < n\}|}{n-1}$
13:             **end if**
14:         **end if**
15:         $convexity\_scores$.append($path\_convexity$)
16:     **end for**
17: **end for**
18: $C \leftarrow \frac{\sum convexity\_scores}{|convexity\_scores|}$
19: **return** $C$

---

In A.1 we provide the procedure with which the graph convexity score can be calculated.

## A.3  Additional results

Figure A.1 shows the correlation between convexity and human-machine alignment split by pretrained and fine-tuned models, as well as by model halves. We observe a high correlation for the first half of all models (R=0.91), while the correlation is lower for the second half of pretrained models (R=0.4) and even negative for fine-tuned models (R=-0.54).

Table A.2 and Table A.3 show the change in OOOA and convexity after optimizing the latent space for alignment (OOOA) using the naive transform [12]. The improvement in convexity after the first and middle transformer block is not as high as after the last transformer block (see Table 1), but still significant, indicating a causal relation between alignment and convexity.

Table A.4-A.6 show the change in convexity after optimizing for OOOA. The positive relation we observed for pretrained models does not hold in this case. For most models, convexity slightly decreases in the middle and last layer while convexity increases in the first layer, which aligns with the negative correlation scores observed in Figure A.1 for late layers in fine-tuned models and positive correlation in early layers.

Since the two scores explored in this paper use different distance metrics to assess the similarity between representations (Euclidean distance in graph convexity and cosine distance in OOOA), we explore whether this discrepancy could influence the results. The correlation coefficient between graph convexity using Euclidean distance and cosine distance is 0.9931 (across all models and layers used in this paper) and 0.9989 for OOOA. Therefore, the influence is negligible.

## A.4  Preliminary results on confounding factors

When comparing the correlations between convexity and alignment on a model basis, there is a notable difference between self-supervised models (BEiT, data2vec) and supervised models (ViT, all fine-tuned models). Self-supervised models show a very high correlation, which decreases drastically with fine-tuning (see Table A.7), while the correlation is lower but stable for the supervised model (ViT). While this hints at a complex relationship, a large-scale study is needed to provide clear conclusions on the role of the objective function.

Generally, we hypothesize that there are multiple factors that could potentially mediate the relationship of human-machine alignment convexity. While we can currently only offer speculations, we hope to provide directions for future work. **Pretraining objective**: It has been suggested in prior work that training objective plays a significant role in determining the human-machine alignment of neural network [7]. A larger-scale study would be necessary to systematically gauge how the training objective influences the *relationship* between convexity and

alignment. We hypothesize that at least the layer where each metric peaks could be affected. Models trained with multi-modal objectives would be another interesting model type to explore, since textual supervision may encourage the formation of conceptual regions. **Fine-tuning**: Fine-tuning is of high practical relevance, given the proliferation of publicly available foundation models. The effect of fine-tuning specifically on human alignment and convexity metrics is as of yet underexplored. Potentially, fine-tuning a model pretrained on a self-supervised task using a supervised objective could exhibit properties on the interpolation between the two. **Weight initialization**: While prior work on alignment and convexity analyzes trends across larger model sets (e.g. [7, 10]), the extent to which parameter initialization affects the learned representations' alignment or convexity has not been determined and could be an interesting direction of future research. **Training dataset size**: All models evaluated in this work have been trained on ImageNet-1k or ImageNet-21k. It may be that the size or diversity of these datasets is not large enough to enable the accurate learning of human-like concepts. Future work could provide an analysis of the effect of training dataset size on the relationship of convexity and alignment to elucidate whether ImageNet variants are sufficient to properly learn the evaluated concepts. Overall, we believe that our work provides a promising starting point for future lines of investigation that may lead to a more integrated view on the human-likeness of neural network representations.

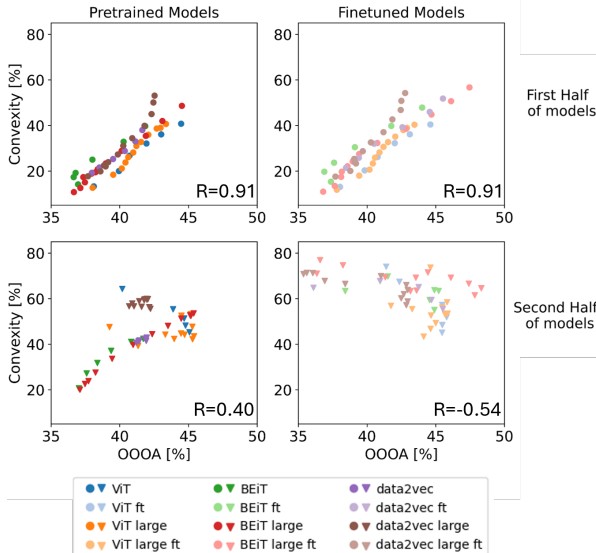

**Figure A.1.** Correlation of OOOA and convexity across models. High correlation in the first half of models (0.91). Lower and even negative correlation in the second half for pretrained (0.4) and fine-tuned models (-0.54) respectively.

**Table A.1.** hugginface.co version of all analyzed models.

| Model | huggingface model |
| --- | --- |
| data2vec base | facebook/data2vec-vision-base |
| data2vec ft | facebook/data2vec-vision-base-ft1k |
| data2vec large | facebook/data2vec-vision-base |
| data2vec large ft | facebook/data2vec-vision-base-ft1k |
| ViT base | google/vit-base-patch16-224-in21k |
| ViT base ft | google/vit-base-patch16-224 |
| ViT large | google/vit-large-patch16-224-in21k |
| ViT large ft | google/vit-large-patch16-224 |
| BEiT base | microsoft/beit-base-patch16-224-pt22k |
| BEiT base ft | microsoft/beit-base-patch16-224-pt22k-ft22k |
| BEiT large | microsoft/beit-large-patch16-224-pt22k |
| BEiT large ft | microsoft/beit-large-patch16-224-pt22k-ft22k |

**Table A.2.** Change in OOOA and convexity after naive transform (first layer). SEM for convexity ±0.1.

| Model | OOOA [%] orig. | OOOA [%] transf. | Convexity [%] orig. | Convexity [%] transf. |
| --- | --- | --- | --- | --- |
| ViT base | 38.1 | +3.9 | 13.6 | +1.7 |
| ViT large | 38.0 | +4.3 | 12.3 | +2.7 |
| BEiT base | 36.9 | +6.0 | 14.3 | +3.2 |
| BEiT large | 36.6 | +2.9 | 10.6 | +2.1 |
| data2vec base | 37.9 | +6.3 | 19.3 | -0.7 |
| data2vec large | 37.6 | +6.2 | 17.6 | +1.0 |
| avg. change (±std) | +4.9 ± 1.4 | | +1.7 ± 1.4 | |

**Table A.3.** Change in OOOA and convexity after naive transform (middle layer). SEM for convexity ±0.1.

| Model | OOOA [%] orig. | OOOA [%] transf. | Convexity [%] orig. | Convexity [%] transf. |
| --- | --- | --- | --- | --- |
| ViT base | 44.4 | +3.9 | 40.9 | -1.9 |
| ViT large | 43.3 | +8.1 | 41.4 | ±0 |
| BEiT base | 41.8 | +10.8 | 39.6 | +1.5 |
| BEiT large | 44.5 | +10.2 | 48.8 | +1.5 |
| data2vec base | 41.4 | +12.4 | 37.2 | +4.2 |
| data2vec large | 42.5 | +12.3 | 52.2 | -3 |
| avg. change (±std) | +9.7 ± 3.2 | | +0.4 ± 2.6 | |

**Table A.4.** Change in OOOA and convexity after naive transform for fine-tuned models (last layer). SEM for convexity ±0.1.

| Model | OOOA [%] orig. | OOOA [%] trans. | Convexity [%] orig. | Convexity [%] trans. |
| --- | --- | --- | --- | --- |
| ViT base | 41.3 | +11.0 | 72.5 | -4.4 |
| ViT large | 44.6 | +7.0 | 68.3 | -2.3 |
| BEiT base | 38.4 | +14.7 | 64.8 | 0.8 |
| BEiT large | 36.3 | +15.8 | 71.7 | -8.1 |
| data2vec base | 33.7 | +19.2 | 66.3 | 0.4 |
| data2vec large | 35.3 | +17.2 | 71.8 | -1.7 |
| avg. change (±std) | +14.1 ± 4.4 | | −2.5 ± 3.3 | |

**Table A.5.** Change in OOOA and convexity after naive transform for fine-tuned models (first layer). SEM for convexity $\pm 0.1$.

| Model | OOOA [%] | | Convexity [%] | |
|---|---|---|---|---|
| | orig. | trans. | orig. | trans. |
| ViT base | 38.0 | +4.0 | 13.4 | +2 |
| ViT large | 37.7 | +4.3 | 12.0 | +2.8 |
| BEiT base | 37.3 | +6.1 | 15.4 | +2.6 |
| BEiT large | 36.8 | +3.0 | 10.4 | +2.0 |
| data2vec base | 38.4 | +6.6 | 20.7 | -1.0 |
| data2vec large | 37.6 | +6.0 | 17.4 | +0.9 |
| avg. change ($\pm std$) | $+5.0 \pm 1.4$ | | $+1.5 \pm 1.4$ | |

**Table A.6.** Change in OOOA and convexity after naive transform for fine-tuned models (middle layer). SEM for convexity $\pm 0.1$.

| Model | OOOA [%] | | Convexity [%] | |
|---|---|---|---|---|
| | orig. | trans. | orig. | trans. |
| ViT base | 44.6 | +8.5 | 41.5 | -1.8 |
| ViT large | 43.4 | +7.8 | 41.9 | -1.3 |
| BEiT base | 44.0 | +9.9 | 47.8 | -1.2 |
| BEiT large | 47.4 | +7.5 | 56.4 | -5 |
| data2vec base | 45.5 | +9.3 | 51 | -0.8 |
| data2vec large | 42.7 | + 11.5 | 52.3 | -3.4 |
| avg. change ($\pm std$) | $+9.1 \pm 1.4$ | | $-2.2 \pm 1.6$ | |

**Table A.7.** Correlation scores between OOOA and convexity on a model basis.

| Model | pretrained | | finetuned | |
|---|---|---|---|---|
| | base | large | base | large |
| ViT | 0.68 | 0.62 | 0.67 | 0.63 |
| BEiT | 0.99 | 0.98 | 0.70 | 0.46 |
| data2vec | 0.98 | 0.95 | 0.44 | 0.27 |

