# OpenReview forum: "Connecting Concept Convexity and Human-Machine Alignment in Deep Neural Networks"
_NLDL.org/2025/Conference — NLDL 2025 Poster_

### Official Review · Reviewer_uXoU · 2024-10-05
**Review of "Connecting Concept Convexity and Human-Machine Alignment in Deep Neural Networks"**

**Confidence:** 3

**Summary:**

The authors empirically study, for various transformer-based vision models, the relationship between the two concepts _human-machine alignment_ and _concept convexity_. For both of these concepts they evaluate an existing proxy/metric, namely _graph convexity_ and the _odd one out accuracy_, and evaluate these on the latent representations of pictures from based classes after various transformer layers in the model. The authors mention that this study provides the first evidence of a relationship between the two concepts. Although it is difficult to draw conclusions from correlation coefficients, the authors also performed an intervention study where they optimize for odd-one-out-accuracy to see the influence on convexity, showing that especially for models that are not finetuned, also the convexity increases.

**Strengths:**

Especially at the high level, I believe the question on the relationship between the representations in machine-learning models and human-machine alignment is interesting.

The article is neatly written, and on a high level the story becomes clear quickly. It is also interesting to study these numbers, and one can probably get more interesting statistics out of them.

It is difficult to analyze correlation scores to interpret relationships, but the authors alleviate this at least partly by doing an intervention study as well: in one of their experiments, they optimize for human-alignment, and measure the effect on convexity.

**Weaknesses:**

In contrast to the larger story, the details were more difficult to figure out for me, and I am still uncomfortable pointing out exactly how various quantities are computed. The authors mention for instance "We performed a correlation analysis of the two scores using Pearson’s R on a layer-wise basis across all models." It was originally unclear to me whether there the number of datapoints in the computation of such a correlation would be just be the twelve averaged scores corresponding to the twelve trained models, or whether there would be a data point for every triplet of classes, for instance. A formula might help, or an even more precise wording.

Similarly, the appendix mentions "The latent representations were extracted according to the way they were used by the classifier in the original implementation. Hence, we took the vector corresponding to the classifier token for ViT and averaged over all the other tokens for data2vec and BEiT." I have difficulty understanding what this means. In general, I am unsure how and where the latent representation of a class is actually computed. This is relevant for the interpretation of the convexity concept.

Although not necessarily a weakness of the paper, it is in a way difficult to connect the proxy to the intended concept, i.e. graph convexity to a measure of convexity, and out-of-order-alignment with human-machine alignment. For instance, whereas the authors in the conclusion talk about the formation of convex regions, it is not necessary for achieving a high correlation score that latent representations of a certain class form a mathematically convex set. On a related, note, the mathematical concept of convexity would not be affected by an affine transformation, while the graph convexity does, because of the construction of the nearest-neighbor graph. The article might benefit from a discussion on these issues.

More detailed remark:
I am wondering why the vectors appearing in the inner product in the definition of Z are not normalized.

**Final Rebuttal Confidence:**

3

**Final Rebuttal Justification:**

The high-level story is interesting to read and it addresses a concrete version of the interesting question how human-machine alignment and latent representations in machine-learning models relate. I also think it is nice that the experiments have been done and that anybody can look at the numbers and ponder about what they mean. That also immediately gives the counterside: it is a a bit difficult to interpret back the results in terms of the higher level question.

I also think the authors responded well to the reviews, by incorporating concerns in the text.

**Justification:**

The high-level story is interesting to read and it addresses a concrete version of the interesting question how human-machine alignment and latent representations in machine-learning models relate. I also think it is nice that the experiments have been done and that anybody can look at the numbers and ponder about what they mean. That also immediately gives the counterside: it is a a bit difficult to interpret back the results in terms of the higher level question.

---

> ### Author Rebuttal · Authors · 2024-10-23
>
> We thank the reviewer for the insightful review and for pointing out some unclarities in our method description. We have updated these parts in the manuscript to clarify all methods and metrics used.
> 1. Pearson's R is calculated on the scores for the convexity and OOOA for each layer by using the scores for each model, so it is indeed based on 12 data pairs for each layer. We added a clarifying sentence in the manuscript.
> 2. We extract the representations after each transformer block, which have the dimensions of 768×197 for base models and 1024×197 for large models. For ViT we use the first embedding patch (the cls token, corresponding to the first row of length 768), which represents the class token of the embedding which is later used by the classifier head. For the other models, we average across the rest of the patch embeddings (resulting in a vector of length 768/1024), as this is used by the classifier head. We updated the description in the appendix.
> 3. The reason the vectors in the Z score are not normalized is that we followed the original work and implementation by Muttenthaler et al. [12].
>
> Regarding the two metrics we use as proxies, we would like to refer to the original publications introducing these metrics [7,10], where a more thorough discussion of these metrics and their properties is made, especially on the properties of the graph convexity during affine transformations [10]. While it is correct that a convex region is not needed to achieve a high correlation or high alignment, we hypothesize that convex regions of human-similar concepts do help alignment, which our results also indicate. We updated the conclusion to clarify this discussion point.
>
> We also added a broader discussion and possible interpretation of our results and possible confounding factors in the appendix, addressing your concern about the difficulties in interpretation. However, as you also very nicely pointed out, this study is meant to start the discussion of connecting several ideas and metrics of human cognition to inform future research in the direction of human-machine alignment and identify new exciting and promising research directions. Therefore, some of the results are still open for interpretation and will require further research to conclusively be interpreted. We present several potential new research directions in the appendix and hope that an open discussion among the community will help identify even more promising research directions and lead to insightful new research in the area of human-machine alignment.
>
> We thank the reviewer for the valuable feedback and pointers to unclarities in the paper. The updated version now contains a more detailed description of the used scores/metrics and a broader discussion on potential research questions and the impact of our work.
>
> [7] L. Muttenthaler, J. Dippel, L. Linhardt, R. A.Vandermeulen, and S. Kornblith. “Human alignment of neural network representations”. In: Conference on Learning Representations (2023).
>
> [10] L. Tětková, T. Brüsch, T. K. Scheidt, F. M. Mager, R. Ø. Aagaard, J. Foldager, T. S. Alstrøm, and L. K. Hansen. “On convex decision regions in deep network representations”. In: ICLR 2024 Workshop on Representational Alignment (2024).
>
> [12] L. Muttenthaler, L. Linhardt, J. Dippel, R. A. Vandermeulen, K. Hermann, A. Lampinen, and S. Kornblith. “Improving neural network representations using human similarity judgments”. In: Advances in Neural Information Processing Systems (2024).

---

### Official Review · Reviewer_Qz4L · 2024-10-08

**Confidence:** 3

**Summary:**

The topic is representation learning. In particular, the paper studies graph convexity and human-machine alignment of the representations learned by neural networks trained either self-supervised or supervised on ImageNet-data. Graph convexity is measured for each class in the THINGS dataset. For human-machine alignment, triplet odd-one-out-accuracy (OOOA) on the THINGS dataset is used. The paper finds that human-machine alignment is typically highest in middle layers of networks, while graph convexity typically increases monotonically with deeper layers. In the authors' words, correlation scores of the two measures over several networks "indicate a significant correlation between these two measures in some scenarios". The paper also studies what happens to the graph convexity when optimizing an affine transformation to increase the OOOA. The graph convexity increases for some networks and decreases for others. The scale of the experiments (12 networks) is ultimately too small to say anything conclusively about the relationship between OOOA and graph convexity. As the authors note, "Further research is warranted to explore under which conditions convexity and human-machine alignment align.".

**Strengths:**

- Results on OOOA and graph convexity individually are interesting.
- The experiments seem correctly conducted.

**Weaknesses:**

- It is not clear why the correlation between graph convexity and OOOA is studied. Is the aim to test Gärdenfors' theory by arguing that if human concepts are convex in neural network latent spaces, then human concepts are convex? If so, what are the conclusions from this study?
- The experiments are too small-scale to give any conclusive results.
- The paper divides the studied networks into "pretrained" and "finetuned", but the networks were pretrained in different ways. As noted in Appendix A.3, perhaps the more important division is self-supervised vs supervised. Again, the number of studied networks is too small to determine this.
- For graph convexity, Euclidean distance is used to build the graph. However, cosine similarity is used to measure similarity in the OOOA measure. This means that if a class has embeddings close to zero, it can be graph convex while having embeddings in all directions from zero, leading automatically to low OOOA. Do the conclusions from the experiments change if the same similarity is used for both measures? One could switch to either use cosine similarity to build the graph for measuring graph convexity or use Euclidean distance to measure similarity in the OOOA.

**Final Rebuttal Confidence:**

3

**Final Rebuttal Justification:**

The authors have addressed the concerns of all the reviewers. The paper contains insights worth discussing at the conference, although the experiments are perhaps not large-scale enough to draw far-reaching conclusions.

**Justification:**

The conclusion of the paper is that there are some situations in which graph convexity and human-machine alignment correlate. This is a quite weak conclusion, but it was impossible to draw stronger conclusions due to the small-scale experiments. Further, the similarities used for graph convexity (Euclidean distance) and OOOA (cosine similarity) are not the same, which could influence results. I believe the paper would be significantly improved if the authors more clearly stated the study's aim and contribution and resolved the mentioned similarity discrepancy.

---

> ### Author Rebuttal · Authors · 2024-10-23
>
> We thank the reviewer for the constructive review and pointing out potential improvements. We would like to comment on the main points:
>
> *It is not clear why the correlation between graph convexity and OOOA is studied?*
> The aim is not to test and verify Gärdenfors' theory, which is widely accepted by the human cognition community, but to investigate whether the convex regions formed by networks align with conceptual regions humans form. In an aligned system we would expect the machine to use the same (at least to some extent) conceptual regions as humans and also group information based on similar features as humans use to distinguish classes. Our experiments show that convex regions formed by the network, that correspond to human-defined concepts, also correlate to some extent with how humans structure information (which we approximate by using the OOOA metric). We added an additional paragraph to the introduction giving a clear motivation of why the two metrics are studied and potential implications of this work.
>
> *The experiments are too small-scale to give any conclusive results. The paper divides the studied networks into "pretrained" and "finetuned", but the networks were pretrained in different ways.*
> We agree and plan to conduct larger-scale studies with additional models and metrics in the future, which will also include a separation of pretraining strategies. We included an extended discussion on confounding factors in the appendix which points out several interesting directions for future research. However, given the novelty of this research area, we believe that initiating the discussion now is valuable. Our findings provide a solid foundation for meaningful dialogue and highlight the potential of this line of inquiry while also giving concrete and informed ideas for future research.
>
> *For graph convexity, Euclidean distance is used to build the graph. However, cosine similarity is used to measure similarity in the OOOA measure.*
> We would like to point out that we are using already established methods that both conducted several experiments with variations of the scores (also based on different distance metrics). For details on the specific methods, we would like to refer to the original publications [7,10]. Our choice to use established metrics is intentional, aligning with existing literature to ensure valid comparisons. While a consistent distance measure might provide slightly different results, our current methodology effectively captures the relationships we aim to study. Nonetheless, we thank the reviewer for the valid concern and agree that this factor should be addressed in future research developing and comparing methods in the area of human-machine alignment.
>
> We thank the reviewer again for the valuable feedback and hope the changes made based on the review highlight the potential of this study and clarify the aim and impact of the presented work.
>
> [7] L. Muttenthaler, J. Dippel, L. Linhardt, R. A.Vandermeulen, and S. Kornblith. “Human alignment of neural network representations”. In: Conference on Learning Representations (2023).
> [10] L. Tětková, T. Brüsch, T. K. Scheidt, F. M. Mager, R. Ø. Aagaard, J. Foldager, T. S. Alstrøm, and L. K. Hansen. “On convex decision regions in deep network representations”. In: ICLR 2024 Workshop on Representational Alignment (2024).

---

### Official Review · Reviewer_hHhT · 2024-10-09
**Connecting Concept Convexity and Human-Machine Alignment in Deep Neural Networks**

**Confidence:** 3

**Summary:**

The paper provides insights into how convexity in latent representations and human-machine alignment are connected. It looks at convexity scores and alignment for 3 different transformer-based networks at different stages/layers using both simply pretrained and fine-tuned versions, as well as how directly improving human-machine alignment via a dedicated transform affects the convexity.
While the results are not entirely consistent across settings, they identify some positive correlation between alignment and convexity and  a potential causal relationship between them.

**Strengths:**

- the paper is well written and provides a clear description of the background work as well as their topic of investigation.
- all the main concepts are well described and illustrated
- it provides useful insight into the relationship between convexity and human-machine alignment, which may inform further works on the topic
(eg. shows that the middle stages of the networks may be more resonable to study in human-machine alignment).
- the code is available and the process well documented

**Weaknesses:**

A few points I would like to be adressed for clarity:
- what is meant by 'centered representations' in 3.3?
- You mention that the convexity and OOOA follow bell-shaped curves. However there are sudden increases again at the last layers for some of the methods. Any comments on that?
- Figure 2C has inadequate caption/description: For which network is that plotted? Since they don't all have the same amount of layers... Is it averaged?
- seems that convexity changes most (for the best) at transformations at first layer, while last layer is more important for OOOA. Any thought on why that could be?
- I find the part on confounding factors (eg. in A.3) lacking/not at all worked out. I would expect you to at least speculate about the topic and further research directions here, instead of simply mentioning the results. After all, this paper is not a method contribution paper, but trying to look under the hood and explain or at least indicate the underlying relationships.

- I'm not entirely convinced the findings on pretrained vs finetuned are generalizable; have you looked at the specific tasks (both for finetuning and pretraining) and how well they resemble the alignment task? The potential problem I see here is that this task for  measuring alignment is biased towards models trained for similar tasks (eg classification of natural images). I would thus be careful when using its results in arguments comparing different networks that were trained for sufficiently different tasks.



Some smaller comments that would further improve the paper:
- you only provide a definition of graph convexity, then describe the convexity score inside text. I would recommend you provide even a formal definition of the score in addition to this.
- legend missing in Figure A1
- more networks and specifically different tasks would be needed to be able to evaluate properly and in a more generalizable way, although I suppose the scope of this paper was really just to present the idea and not a thorough evaluation.
- mention the existence of additional results (specifically the results from other than last layers)  that are in the supplementary clearly in main text

**Final Rebuttal Confidence:**

3

**Final Rebuttal Justification:**

The authors have further improved the paper according to the reviews. All the comments point to the fact that while the experiments (or argumentation for some of the choices made) are not the strongest, the presented idea and insights are interesting for the wider community.

**Justification:**

There are of course points that could be improved and the paper as it is provides only an incremental improvement in the understanding of the interplay between convexity and alignment (could be improved with more experiments/evaluations and targeted discussion). However, I think that it is definitely of high enough quality in terms of writing and, more importantly, illustrates a new way of looking at convexity&human-machine alignment (and ultimately explainability) that is worth sharing more broadly and could very well inform and direct further research on this.

---

> ### Author Rebuttal · Authors · 2024-10-23
>
> We thank the reviewer for the comments and helpful pointers for improvement. In the following, we would like to add some clarifications:
>
> *What do “centered” representations mean?* Instead of the original extracted representations for measuring alignment, we first subtract the mean representation and divide them by the standard deviation. This follows the prior literature (Muttenthaler et al. 2023). We added a clarifying statement in the Background section. We thank the reviewer for pointing out this missing detail.
>
> *Is there an explanation for the sudden increases in convexity in the last layer?* Admittedly, we do not have a good explanation for this. This effect can mainly be observed for ViT large, but not for other models. We hope that future studies investigating more factors, like pretraining strategy, can shed more light on this.
>
> *Which networks are depicted in Fig 2C?* We thank the reviewer for this pointer and improved the clarity of the caption. The line is indeed calculated over all the networks, i.e. it is the correlation of the OOOA of all networks at a given layer and their convexity at the same layer. As the correlation is already some form of aggregation, no further averaging was needed.  For layers with higher layer count, we selected the layers that lie closest to the fewer-layer model layers, in terms of the ratio layer-position/last-layer-position.
>
> *The transform increases convexity in the first but OOOA at the last layers most strongly - why?* We hypothesize that the concepts we evaluate convexity for and the concepts that most influence human-machine alignment build on the same set of lower-abstraction-level concepts. As those lower-level concepts may be best decodable from earlier layers, optimizing OOOA there also increases convexity. At a higher abstraction level, the concepts for which convexity was evaluated and the ones humans used to solve the triplet task may diverge. Since these concepts may be more complicated concepts, building up on the shared early-layer primitives, optimizing OOOA at higher layers is more effective, but also coincides less with convexity.
>
> *What more can be said about confounding factors?* We expanded the corresponding section in Appendix A. 4 to provide a more comprehensive view on possible confounding factors and some pointers to interesting directions of future research.
>
> *What role does the training objective play when comparing pretrained and fine-tuned models?* We agree with the reviewer that, given the results presented in the paper, the difference in the representation measures between pretrained and fine-tuned models is not clear-cut. As we hypothesized in the paper, the training objective is likely to play a significant role in how concepts are represented and in how convexity and alignment relate to each other. The fine-tuning task, as well as the pretraining for the ViT model, are classification-based, whereas the pretraining for the other models is self-supervised. Generally, the training objective has been shown to be a significant factor for OOOA but classification does not seem to lead to particularly well-aligned models (see Muttenthaler et al. 2023). Given the practical relevance that fine-tuning has in the use of publicly available models, we think it is valuable to pay special attention to it, but we agree that a larger study, where variations in the pretraining/fine-tuning objectives are isolated, would be desirable.
>
> *Regarding the minor comments:* a) We opted for a prose description of the calculation of the convexity score since it is more algorithmic in nature, and a purely mathematical formulation would be quite convoluted. To add clarity, we added an Algorithm box in the appendix that more succinctly describes the procedure. b) We added a legend to Figure A1. c) We agree with the reviewer that a larger scale study would be highly interesting for future work, but we found that even at this scale, the results would already be interesting to the community and hopefully contribute to fruitful discours. d) We added references to the appendix in Sec. 4.2 to point to the existence of additional results.
>
> We would like to thank the reviewer for their constructive comments and believe that the changes made in response have improved the clarity and overall quality of the paper.

---

### Official Review · Reviewer_95m5 · 2024-10-12
**Interesting unusual metrics with potential benefits**

**Confidence:** 4

**Summary:**

The paper describes a novel approach of measuring convexity in neural networks and relating it to the alignment with humans, by measuring the performance on the OOOA (odd one out task). The study demonstrates that convexity in neural networks seem to be related with human alignment in the way information is organized in the latent space. They show that optimizing for OOOA task have a positive effect in convexity in Neural networks.

**Strengths:**

The paper provides an interesting and novel method to study neural networks and the alignment with humans with respect to the out one odd task.

**Weaknesses:**

I might have missed it, but I feel that while the concept of convexity is nicely introduced,  a more clear understanding on why is it desirable is missing? What do we get out models that are more convex? Are they more interpretable?  The authors mention briefly in the intro that convex systems are more robust, but there is no evidence presented in the paper about this. Could a robust study be included to show how the increased convexity helped the models after the alignment was provided?

Also not sure why there are no mention on any convolutional models that have been shown more aligned with humans in terms of architecture.

**Final Rebuttal Confidence:**

4

**Final Rebuttal Justification:**

Given the positive reviews that other reviewers have raised. I upgraded my score.

**Justification:**

I think there is novelty in including the convexity as a measurement of capacity for models, but it seems the results are half way through, the authors show that increasing OOOA accuracy helps increasing the convexity, but what happen next? Perhaps more work into showing this benefit would make the paper stronger and a clearer higher  impact to the community.

---

> ### Author Rebuttal · Authors · 2024-10-23
>
> We thank the reviewer for the comments and helpful pointers for improvement. In the following, we would like to add some clarifications:
>
> The graph convexity score was introduced and defined in reference [10]. This paper also provides a justification for why convexity is an interesting and potentially useful property. Most of the motivation originates in cognitive science. Works from Peter Gärdenfors and other cognitive scientists suggest that humans represent concepts as convex spaces. Convexity is also claimed to support robust generalization, few-shot learning, and communication. Moreover, one of the findings in [10] is a relation between the convexity of classes in pretrained models and the accuracy of that class after fine-tuning. We take this result as a motivation for why increasing convexity is desirable. We extended the motivation by adding an additional paragraph to the introduction section.
>
> Since we explore how the convexity and OOOA evolve throughout the network, transformer architectures are the most convenient because embedding spaces of the same dimension can be naturally extracted after each transformer block and compared directly. When using convolutional networks, more exploration of how exactly to extract the features in a way that is comparable throughout the whole network is needed and will be part of future work to extend our analysis to CNNs.
>
> We agree that a more thorough study including multiple network architectures (including convolutional models), training strategies and metrics is needed and we plan to conduct it in the future. However, given the novelty of this research area, we believe that initiating the discussion now is valuable. Our findings provide a solid foundation for meaningful dialogue and highlight the potential of this line of inquiry while also giving concrete and informed ideas for future research. We added an additional paragraph to Appendix A.4 discussing potential confounding factors and ideas for future studies.
>
> [10] L. Tětková, T. Brüsch, T. K. Scheidt, F. M. Mager, R. Ø. Aagaard, J. Foldager, T. S. Alstrøm, and L. K. Hansen. “On convex decision regions in deep network representations”. In: ICLR 2024 Workshop on Representational Alignment (2024).

---

### Meta-Review · Area_Chair_RniS · 2024-11-04

**Recommendation:** Accept (Poster)
**Confidence:** 4

**Metareview:**

The paper examines the relationship between convexity in the representations produced by neural networks and human-machine alignment. They establish a correlation in real-world models between these quantities. Additionally, they find that convexity tend to increase when optimizing for alignment.

The reviewers agree that the topic of the paper is interesting and provides insights worth publication. Given the nature of the reviews and that the authors have improved the paper by addressing the concerns raised by the reviewers during the rebuttal phase, I recommend accepting the paper.

**Suggested Changes To The Recommendation:**

1: I agree that the recommendation could be moved down

---

### Decision · Program_Chairs · 2024-11-06

**Decision:**

Accept (Poster)

**Comment:**

We recommend a poster presentation given the AC and reviewers recommendations.